# School Food Environment in Urban Zambia: A Qualitative Analysis of Drivers of Adolescent Food Choices and Their Policy Implications

**DOI:** 10.3390/ijerph19127460

**Published:** 2022-06-17

**Authors:** Mulenga Mary Mukanu, Anne Marie Thow, Peter Delobelle, Zandile June-Rose Mchiza

**Affiliations:** 1School of Public Health, University of the Western Cape, Bellville 7535, South Africa; zandile.mchiza@mrc.ac.za or; 2Menzies Centre for Health Policy and Economics, University of Sydney, Camperdown, NSW 2006, Australia; annemarie.thow@sydney.edu.au; 3Chronic Disease Initiative for Africa, University of Cape Town, Cape Town 7700, South Africa; pdelobelle@gmail.com; 4Department of Public Health, Vrije Universiteit Brussel, 1090 Brussel, Belgium; 5Non-Communicable Diseases Research Unit, South African Medical Research Council, Cape Town 7505, South Africa

**Keywords:** policy interventions, school food environments, adolescents, food choice

## Abstract

Identifying context specific points for reforming policy to promote healthier food environments and consumer behavior in critical life stages like adolescence is crucial in addressing the double burden of malnutrition. Using a qualitative study design, we conducted 20 focus group discussions with grade 10 pupils from ten secondary schools in Lusaka. Turner’s framework which conceptualizes the food environment into two domains—the external domain (availability, pricing, vendor and product properties, and marketing and regulation of food) and the internal domain (accessibility, affordability, convenience, and desirability of food)—was used to guide thematic data analysis and results interpretation. Adolescents stated their food choices are largely based on personal preference linked to the need for social acceptability among peers. Adolescents felt their food choice is limited to ‘cheap junk foods’ which are affordable and readily available at school. Healthy foods like fruits were said to be inaccessible and unaffordable by the majority of adolescents. Some adolescents stated they were attracted to certain foods by adverts they see on TV and social media. School food environments in urban Lusaka do not support healthy food choices. Policy reforms are required to increase access to affordable healthy food options in schools, and to curb the indiscriminate marketing of unhealthy foods to adolescents.

## 1. Introduction

Adolescence is one of the critical life stages as behaviors and habits developed during this formative period are likely to be carried into adulthood [1]. If these behaviors are not monitored, and effective policy interventions developed, this period of life can be the beginning of negative health outcomes like obesity. The World Health Organization classifies adolescents as those aged between 10 to 19 years [2]. Therefore, understanding the factors shaping food choices during adolescence is important as this may provide a point of entry for targeted and cost-effective interventions aimed at addressing the double burden of malnutrition.

Social cognitive factors at individual, community and societal level have been shown to influence food choices of adolescents in many contexts [3,4]. At individual level, consumption of snacks and fast foods, especially among adolescent girls, is desirable due to peer influence, taste, and appearance of food [5]. Adolescents concerns about food safety and hygiene also increases the desirability of snack foods which are usually conveniently packaged [6]. Family, cultural norms and expectations constitute some of the community level influencers of food choice as they influence food production, food selection, food preparation, and consequently food consumption and overall nutritional outcomes [7,8,9]. For example, improved diet quality has been demonstrated in adolescents who eat meals with their parents [10]. Media, as a channel of marketing food and beverages to children remains one of the dominant societal influencers of food choice. For instance, there is growing evidence to suggest that adolescents who are regularly exposed to unhealthy food advertisements on television are more likely to choose and consume fast foods and soft drinks [11,12]. In addition, the adolescents’ high use of smartphones and social media platforms exposes them to predatory food marketing where unhealthy foods are portrayed as cool and exciting [13,14].

Of particular importance in the understanding of food choices of adolescents is the school food environment. The school food environment is defined as the ‘spaces, infrastructure and conditions within and beyond the school premises where food is available, obtained or purchased and consumed’ [15]. This includes the food information available within school premises as well as promotion and pricing of foods and food products [15]. Having unhealthy food in and around school premises has been associated with increased likelihood of children consuming such food in many settings [16,17]. Such evidence coupled with statistics of the growing epidemic of obesity in children [18] calls for deliberate efforts to build school environments that support healthy food consumption behaviors.

Consumer food choices, demand and habit formation are affected by food environments [16,19,20,21]. The food environment has been defined as the ‘physical, economic, political and socio-cultural context in which consumers engage with the food system to make their decisions about acquiring, preparing and consuming food’ [20]. As such, food system research advocates for policy measures that influence availability, desirability, and accessibility of different food types to promote healthier dietary patterns. Experts suggest that policies relating to food composition, labeling, promotion, provisioning, retailing, pricing, trade, and investment are required to make food environments healthier [20]. Similarly, the WHO recommends that schools and other settings for children implement comprehensive interventions to create healthy school food environments that will prevent the double burden of malnutrition and NCDs [22]. Establishing standards of foods and beverages sold in schools to ensure that they meet healthy nutrition guidelines, eliminating the provision or sale of unhealthy foods in the school environment through zoning of food vendors and inclusion of nutrition and health education within the core curriculum in schools are policy measures that are showing promise of making the school food environment healthier [14,22]. However, the adoption and implementation of such policy measures remains low in many African contexts [23,24,25].

The growing double burden of malnutrition is an emerging public health concern in Zambia. Early evidence shows that Zambia is undergoing a nutrition transition with a shift from the traditional Zambian diet of *nshima* (a thick porridge made from maize meal) (which is consumed with a variety of protein and/or vegetable relish) to more processed fast foods [26]. Consequently, nearly a quarter of the adult population are overweight or obese [27] while 35% of children under five years are stunted [28]. Data on the nutrition outcomes of adolescents is largely lacking and policy gaps relating to adolescent nutrition still remain despite nutrition being a policy priority in Zambia. Our unpublished data indicate that regulations restricting marketing of unhealthy food to children and standards for healthy provisioning of food in the school food environment are absent in Zambia [29].

Despite the importance of adolescent nutrition, food environments as key determinants of food choices remain an understudied area in the African context, especially in Zambia. Hence, this study used a food environment lens [30] as follows: (1) to map the school food environment of adolescents; (2) to identify the drivers underpinning food choice among school going adolescents in Zambia and (3) to analyze the implication of these outcomes on food policy. Such an approach is best placed to identify context specific points of intervention for reforming policy to promote healthier food environments and consumer behaviors.

## 2. Materials and Methods

### 2.1. Study Design

A qualitative research design was used for this study. We conducted structured observations of the school food environment of selected secondary schools in Lusaka District and focus group discussions [FGDs] to obtain an in-depth understanding of factors that influence dietary patterns of school-going adolescents. The findings from the school food environment mapping and FGDs were then integrated, and their implications for policy examined.

#### Conceptual Framework

We applied the conceptual framework for exploring food environments in low to middle income countries (LMICs) proposed by Turner and colleagues [31]. This work places the food environment within the broader food system and describes it as an interface where people interact with the wider food system to acquire and consume food. In understanding the food environment from the consumers’ perspective, Turner and colleagues (2018) categorize dimensions that mediate acquisition and consumption of food into two domains: external (availability, price, vendor and product properties and marketing and regulation of food) and internal (affordability, convenience, accessibility and desirability of food). It is within these domains that we explored the key influencers of adolescents’ food choices through FGDs.

### 2.2. Study Sites

The study was conducted in Lusaka District in Lusaka, the capital city of Zambia. Lusaka District has 4 main types of residential areas based on the population density: (1) unplanned high-density residential areas, (2) unplanned low density residential areas, (3) planned medium-high density residential area and, (4) planned low density residential areas [32] For the purpose of this study, the status of whether a residential area is planned or not was not considered and consequently unplanned and planned low residential areas were merged into one category. The population density of the neighborhood a school was located in was used as a proxy for the general socio-economic status of the learners attending that school, an approach which has been used in other studies [33].

The study was conducted in 10 purposively sampled secondary schools in urban Lusaka of which 9 were government funded and 1 was privately owned. The purposive sampling of the schools was done to ensure diversity in terms of socio-economic status and ownership. According to a list of secondary schools obtained from the Ministry of Education, they were at the time of this research, a total of 54 government and 7 private owned schools in Lusaka district. From these schools at least 20% from both the government- and private owned schools were selected. Table 1 shows the characteristics of the selected schools included in the study. Two private schools were approached for participation; however, permission was only granted by one school. Permission was granted by nine government owned schools, with one low density residential school denying participation.

### 2.3. Study Tools

A semi-structured guide for the FGDs was developed based on the eight dimensions from Turner’s conceptual framework. The questions assessed how the dimensions of the food environment impacted the food choices of school going adolescents (see Appendix A). The tool was pilot tested through two FGDs at a school with similar characteristics to the schools included in the sample.

A semi-structured observation checklist was developed and used to conduct the school food environment mapping (see Appendix A). The observations belong to a qualitative research methodology where a phenomenon is observed in its natural setting and recorded by a researcher [34,35,36,37]. The mapping checklist was developed to collect information on the typology and location of food vendors in and around all the sampled school premises as well as the type of food sold, based on the external domains of the Turner’s conceptual framework. The checklist was completed by a research assistant.

### 2.4. Sampling and Data Collection

Participants in the FGDs were randomly selected from Grade 10 learners who completed the food frequency questionnaire in the larger quantitative study. Authorization to conduct the study was sought from the Ministry of General Education. Permission was sought from the school administration before any study activities could be conducted by presenting the authorization letter from the Ministry of General Education to the school administrator of a selected school. Once the school granted permission, the trained research assistant obtained the sampling frame, consisting of names of all grade 10 learners (available on that day), from the teacher assigned by the school administrator to oversee the study. Random sampling was then conducted by the main author to select a random sample of 40 learners in each school to whom the food frequency questionnaire was administered. Each learner was assigned a unique random number and the numbers were then sorted from smallest to largest. The first 40 learners on the list were invited to complete a food frequency question after which they were also invited to participate in the FGD the following day. The selected learners were given information sheets and parent permission forms for their guardian to complete before they participated in any study activities. Learners had to further assent to participate in the study by completing an assent form, and they were informed that they still had the right to choose not to participate with no negative consequences for them.

A total of 20 FGDs with adolescents (two from each of the ten schools) were conducted. Each FGD had an average of 10 learners, resulting in an overall 200 participants. The discussions were conducted after classes within the school premises in a quiet environment such as the school hall or the sports field. The FGDs lasted an average of one hour forty minutes. Learners participating in the FGDs were provided with refreshments as required by the local ethics review board. Data collection was conducted in November 2020.

All data collection was conducted by trained research assistants who were supervised by the lead author of the current manuscript. A total of five research assistants were used to collect the data. All research assistants have a minimum of a bachelor’s degree and have had at least one year experience collecting qualitative data. The research assistants were trained on ethical conduct of research where justice, beneficence and right of person was emphasized. Research assistant training also included an introduction to qualitative research methods and principles of collecting qualitative data as well as an introduction to dietary assessment in adolescents.

### 2.5. Data Analysis

#### 2.5.1. Focus Group Discussions

Thematic analysis using a hybrid approach, including deductive framework analysis and inductive theme development [38] was used for the analysis of FGDs. We developed a set of predetermined themes based on the study’s conceptual framework [31], as well as emergent themes arising from the data; these were mainly related to broader, context specific understanding of the drivers of dietary patterns. The hybrid thematic analysis was conducted manually in five stages [38] and included a reflexive process. First, FGD transcripts were read for familiarization and understanding. Second, framework analysis was conducted where data was coded to the eight ‘overarching’ a priori themes based on the aforementioned conceptual framework. Third, data coded to these ‘overarching’ a priori themes (outlined in Table 1) were analyzed to inductively identify codes which constituted key attributes of food or food environments that adolescents felt influence food choice. These codes were then grouped into subthemes (where possible) under each overarching theme. This process was repeated until data saturation was reached where no new attributes were identified from the data. Fourth, the identified codes were reviewed to check for divergent attributes that might have not been captured by the a priori themes. Furthermore, subthemes under each overarching theme were also reviewed to ensure they correctly captured the attributes under them. In the fifth step, each theme and accompanying subthemes were analyzed to identify the common thread and were organized in a way that best reflected the data. The codes applied to data from FGDs under each main theme are summarized in Table 2. For quality assurance the coding and analysis processes were independently checked and approved by the co-authors of the current manuscript.

#### 2.5.2. School Environment Mapping

The structured observations of the school food environment were summarized, and descriptive statistics computed in the form of frequencies and proportions using Stata version 15 [39] statistical analysis software. Findings from both the FGDs and school food environment mapping were synthesized and integrated in the discussion of policy implications.

## 3. Results

In this section we present a description of the school food environment mapping as well as the adolescents’ perceptions of the drivers of their food choices obtained from the FGDs. We adopted an integrated approach to understand the adolescent’s perceived drivers of food choices in the context of their corresponding school food environment in order to provide a comprehensive discussion of policy implications.

### 3.1. Description of Food Sold in the School Environment

In all the 10 schools there were tuck shops within the school premises and other food vendors outside the schools. Food vendors outside the school premises included street vendors who are mobile or operate from semi-permanent structures locally called *kantemba* as well as groceries, supermarkets, and convenience shops. Mobile street vendors were mainly observed stationed at the entrance of the school while semi-permanent structures were often set up on busy street corners near the school. Groceries and supermarkets often carried food and sugar sweetened beverage advertisements. Healthy foods including fruits and traditional snacks like roasted maize were observed to be mainly available from street vendors.

The types of food we observed being sold in the school food environment are presented in Figure 1, summarized by availability in school tuck-shops versus other food vendors outside the school. Snacks—i.e., biscuits, chocolates, *jiggies* (corn-based processed packaged snacks), *freezits* (sugar sweetened drink sold frozen), potato crisps, sugar sweetened beverages (SSBs)—and water were available in 100% of the schools and were sold in both the school tuck shops and by other vendors outside the school. Milk and fruits were more readily available with other vendors than in school tuck-shops. Only 10% of the schools had tuck-shops selling milk and fruit. In comparison, 90% of schools had other vendors selling fruit and 30% had milk available. However, fast foods (i.e., fried chicken, chips, sausages, pies, shawarmas) were more available with other vendors than in school tuck shops.

### 3.2. Focus Group Participants’ Characteristics

A total of 200 Grade 10 learners participated in the FGDs. Only one FGD was conducted with learners from a private school. Of the 19 FGDs conducted with learners in government funded schools, seven of these were in low density neighborhood schools while six were in middle and six in high density neighborhood schools. The majority of these learners were females (68%) while 32% were males.

### 3.3. Drivers of Food Choice

In the sections below we present the identified key drivers of food choices based on the domains outlined in the conceptual framework of Turner et al. [28].

#### 3.3.1. Internal Drivers of Food Choice

Desirability

Personal preference was identified by adolescents as a strong determinant of what they choose to eat. Adolescents largely linked their desire/food preference to taste and appearance of the food items. Traditional and ‘healthy’ food was generally seen as less exciting and visually unappealing. Adolescents stated that they usually consume traditional foods for inherent health benefits. Some adolescents reported unhealthy habits such as adding more salt as a way to improve the palatability of some traditional food eaten at home. The majority of adolescents stated that they desired fast food like *Hungry Lion* (an international franchise selling fried chicken and chips) because such food looks appealing and tastes delicious.

Food choice was linked to the need for expression of autonomy by young people. Most adolescents felt that eating outside the home was an opportunity to break the culture of the traditional staple diet (*nshima* and relish) eaten at home, in which they usually have no choice in deciding. Adolescents stated that they felt happy when they bought and ate food of their choice.

“It is empowering in the sense that I now have a choice. I can make a choice to buy what I want, something healthy or something junky, something tasty, something sweet, I can make a choice.” Female, 15 years, Low density neighborhood school.

Social acceptability contributed to the desire for particular food items. The majority of adolescents said they favor franchise fast food because it is what is trendy among their peers and they do not want to be left behind. While healthy food consumption was generally frowned upon, some adolescents saw it as an opportunity to be trendy and unique.


*“Purchasing healthy food is also nice because you get to standout because most people like to eat junk food and when you are just eating healthy food you look unique and it’s just nice when you feel good about yourself.” Female, 15 years, Low density neighborhood school.*


Adolescents related food choices to nutritional knowledge. The majority of adolescents had knowledge of the negative implications of consuming certain foods. ‘Modern foods’ were associated with development of ill health. For instance, consuming sweets, chocolate, and carbonated drinks was linked to dental carries and ‘sugar disease’ (i.e., diabetes) while fatty foods were linked to the development of obesity and hypertension. In addition, adolescents were able to identify the nutritional benefits of different food groups. Adolescents explained that fruits and vegetables provide nutrients and vitamins required for good growth and carbohydrates provide energy while snacks provide no nutritional value to the body. Vegetables like *bondwe* (a species of amaranthus), spinach and beetroot were also identified for their benefits of blood building. A few adolescents added that the food preparation method and quantity consumed also determine how healthy the food will be.

However, adolescents also had misconceptions about food. The perception that healthy food is only eaten at home or when one is unwell was common among the adolescents. Some adolescents associated the cost of food to its ‘healthiness’. The relatively expensive fast foods like *shawarma* (a flat bread-based wrap filled with chips, meat, and sauce)*,* burgers and pizza were viewed as healthy by some adolescents. Overall, adolescents felt that healthy food consumption is hindered by lack of knowledge on what constitutes healthy food and lack of skills for healthy food preparation.


*“Mostly people my age are not actually aware of what healthy foods they need to eat because mostly parents or guardians are usually busy and they don’t have time for us to educate us on what we are supposed to eat” Female, 16 years, Medium density neighborhood school.*


Affordability

Adolescents in the FGDs reported that their choice of what to eat is largely determined by their purchasing power. Adolescents felt that their purchasing power as school going learners was low since they do not have a source of income, and this only allows them to afford cheap food like snacks. Adolescents identified franchise fast foods like *Hungry Lion* and *Debonairs Pizza* among the foods they are unable to purchase. Adolescents unanimously agreed that healthy food options were too expensive for them to afford. They felt that they would eat healthier food options like fruits if this was more affordable as well as available within their school food environment.

Accessibility

Lack of accessibility was one of the barriers to consuming healthy food identified by adolescents. The majority of adolescents stated that vendors that sell healthy foods like fruits were a distance from the school and they were not allowed to leave school premises during school hours. As a result, they are ‘forced’ to eat junk food because vendors who sell such foods are readily accessible within the school (tuck-shop) and outside (*Kantemba* and street vendors). Our observation of the school food environments shows that street vendors selling fruit were often found on busy street corners near the schools and only accessible to learners after school hours, supporting the assertion by adolescents.

Technological and developmental advances were also associated with the shift in dietary patterns of young people compared to their parents. Adolescents attributed the desire and increased consumption of fast foods among their age group to the accessibility of these foods in the community. They explained that the coming of malls and delivery services offered by fast food companies are making fast food ubiquitous.


*“When they [parents] were younger I think the availability of junk foods wasn’t as common as it is here for us. For them it was harder and more expensive to find junk foods like burgers unlike foods that is more traditional. For us you can just go anywhere and you find a Hungry Lion store”. Male participant, 17 years, High density neighborhood school.*


Convenience

Food that is easier to prepare and consume appeals to young people. The majority of adolescents stated that in the home setting, they usually feel lazy about preparing food that takes a lot of skill and time such as *nshima* and other healthy foods. They explained that healthy food also needs a lot of ingredients which are not readily available. Noodles, eggs, bread, sausage, and cereal were listed as the common foods prepared as they require only a few minutes to be ready. Some adolescents added that their preferred cooking method is frying because it is fast and the food tastes good when fried and crisp.

#### 3.3.2. External Drivers of Food Choice

Price

Price of food items determines the type of food young people will eat. All the adolescents agreed that they were drawn to buy and eat snacks and junk food because the price of these food items was less than that of more expensive food options. The commonly consumed food items bought in and around the school (such as *jiggies, freezits*, fritters, cookies, and popcorn) were reported to cost less than K2 (US$0.08). Fruits such as apples were generally considered expensive by adolescents. They stated that on average, a banana costs K2 (US$0.08), an apple costs K4 (US$ 0.16) while an orange costs K8 (US$0.32).


*“Healthy foods are very expensive when you compare them to these junky foods that we like to buy. For example, one apple is K3 and then I think if am going to buy Jiggies which are K0.50, I am going to have maybe about six! So I’ll decide to buy those ones [jiggies] which I am going to have more rather than buying the apple” Male, 15 years, Medium density neighborhood school.*


Availability

Adolescents used the term ‘snacks’ and ‘junk food’ to classify the food available in the tuck shop at their school. They explained that the availability of cheap snacks responds mainly to their purchasing power as adolescents as well as their tastes and preferences as young people. Adolescents from two schools mentioned availability of fruits such as mangoes, oranges, and guavas on the school premises where they get fruits when in season. Our school food environment observation also found wide availability of snacks, fast foods, and SSBs being sold by vendors in and around all the schools.

Adolescents unanimously felt that the food available in the school tuck shops lacked variety and there was no healthy food in stock. However, when asked about additional food that should be stocked in tuck shops, adolescents still chose fast food options like fried chicken and chips, pizza, and *shawarma*. Only a few adolescents felt the school tuck shops also needed to have fruits and pure fruit juices, smoothies and salads, which they categorized as ‘healthy’, in order to cater for those who are health conscious.


*“Personally, I would like them to sell fruits in school because some of us are just gaining weight at school because of the junk foods they are selling.” Female, 14 years, Low density neighborhood school.*


Marketing

Adolescents reported seeing adverts for food and beverages frequently on TV. Other advertising channels mentioned were radio, internet social media platforms, billboards, newspapers, and catalogues produced by supermarkets. Food and beverage adverts were also observed on store fronts of formal food vendors such as groceries in the school food environment.

Social media marketing especially was an influencer of food choices and was linked to social acceptability. Adolescents said that they often learn about trending eating places, or the foods they should eat, from posts on social media platforms such as Facebook, Instagram, and YouTube. Some adolescents said they follow the dietary pattern of favorite celebrities so that they can have a similar body image.


*“As for me, let me give an example of tamarind: I heard Kim Kardashian on that show [Keeping up with the Kardashians] say that for her body to be like that, she eats tamarind. If she eats that, even us to keep our bodies the way she looks, now we start eating tamarind” Female, 14 years, Medium density neighborhood school.*


Zambian food manufacturers of SSBs engaged in marketing activities that directly targeted young people. Adolescents explained that these Zambian companies use advertising tactics such as road shows which are appealing to young people when they want to introduce a new drink, so that young people know about their product and buy it.


*“I have seen road shows… It’s when they bring trucks with music such that when you go up on the platform in the truck and dance; you are given mojo [a brand of carbonated drink] so that know that there are drinks like mojo.” Male, 17 Years, Medium density neighborhood school.*


Vendor and product properties

Adolescents considered the hygiene of food vendors when deciding where to source food from in addition to the pricing of food items. All the adolescents were conscious that buying food from a dirty environment is associated with a risk of diarrheal diseases and COVID-19. Other attributes considered during vendor selection were the popularity of the vendor and facilities for in-house dining. However, the majority of adolescents added that while they are aware of the standard that food vendors are supposed to maintain, where they buy food is ultimately determined by the price as they have very low purchasing power. Some adolescents further added that they consider the packaging of a food item as it tells a lot about the quality of the product and the standards of the manufacturer.


*“We look at the quality, the way products are looking. This is important because I cannot buy food anyhow; I have to buy food that is profitable to my health, not eating because you just want to eat. Nowadays there are a lot of cheap things that are of low quality.” Female, 15, High density neighborhood school.*


## 4. Discussion

Our research aimed to map school food environments and identify the food environmental factors influencing food choices of school going adolescents in Zambia. We found that the school food environments do not adequately support healthy food choices with unhealthy food commonly sold in and around the school with limited access to healthier options. We also found that the influential drivers of food choice included desirability and affordability in the internal domain, and availability and marketing in the external domain of the conceptual framework of Turner et al.

As this is the first study in Zambia to document adolescent engagement with school food environments, these findings provide insights for Zambian food and nutritional policy. The discussion thus focuses on the policy implication of our findings in the light of international and evidence based best practices for improving food environment and promoting healthier food choices among school-going adolescents.

### 4.1. Improving Availability of Healthy Food

Our study findings suggest that school food environments in urban Lusaka are largely unhealthy owing to the wide availability of cheap ultra-processed snacks like biscuits, corn-based snacks, sweets, and sugar sweetened beverages. A similar profile of school food environment has been shown in other African countries [40,41]. Interventions that will support availability and accessibility of healthier food in the school food environment of urban schools in Zambia are therefore required.

Policy interventions targeted at food provisioning and retailing within and around schools have the potential to promote the availability of healthy food in the school environment [42,43,44,45]. Healthy food provisioning in schools can be achieved through the introduction of school feeding programs, school gardens, and healthy food supplies in school tuck shops/cafeteria [8,14]. Availability of healthy food in the school food environment results in relatively higher consumption of healthy foods like fruits [46]. However, a South African study showed that older adolescents may not appreciate healthy food provisioning as this is seen as infringing on their autonomy on food choices [47]. The Ministry of Education in Zambia through the School Health and Nutrition policy requires school tuck shops/cafeteria to sell healthy food as one of the measures of improving the nutritional status of learners [48]. However, it is clear that this policy measure is not enforced as shown from our finding where only 1 out of 10 schools in our study sold healthy food. This is not a unique situation as poor implementation of school food policies has been shown in other settings in LMICs [23,49,50].

Recent evidence from LMICs in Africa and Asia implicates the density of unhealthy food outlets in an environment to poor dietary patterns and health outcomes [51,52,53]. Zoning laws that work to limit the spatial distribution of unhealthy food outlets could be of use in Zambia, given the outcomes we presented that suggest a high density of food vendors selling unhealthy food in the Zambian schools. Such initiatives of zoning food outlets around schools have been successfully implemented in some high-income country communities, especially in the United States of America [52]. These initiatives are supported by the powers granted to municipal authorities. While zoning laws have shown good promise in high income settings, such policies might have limited applicability in LMICs as they require enforcement and monitoring, which are important policy making components that are weak in most LMICs including Zambia [23,24,25,54,55]. Our findings also indicate that informal food vendors are important players in healthy food provisioning in the school food environment as they were found to be the main sellers of fruit in the current study. This finding is corroborated by the findings of Steyn, Labadarios and Nel [56], where they showed that fruit was the most commonly purchased street food in South African schools. Designing of zoning laws in Zambia will therefore need to be nuanced in order to support informal vendors’ healthy food provisioning in a safe environment.

Additional supportive policies from other sectors dealing with community development, agriculture and water and sanitation are also required to promote availability of healthy foods. Our study also shows that fruit trees within school premises are a source of fruits for adolescents. The Zambian agriculture sector can thus support planting of fruit trees in and around schools to improve seasonal availability of home-grown local fruits. Evidence from low-income settings in Uganda links the availability of fruit trees to improved nutritional outcomes of children [57].

### 4.2. Improving Affordability of Healthy Food

Adolescents in our study generally felt they had limited purchasing power, hence they could not afford healthy food. A quarter of adolescents from our sampled schools reported not having pocket money while 75% reported receiving pocket money that amounted to an average of $0.50 (K10) [58]. Policy interventions are therefore required to address the price and affordability of healthy foods which currently might be restrictive for an average school going adolescent.

Price related polices have widely been used as a measure to regulate affordability of both healthy and unhealthy foods. Sugar sweetened beverage taxation is one of the recommended policy measures which aims to increase the price and thus curtail the consumption of sugary drinks linked to the rising obesity burden [59]. Reduced consumption of sugar sweetened drinks has been recorded in countries like Mexico, the UK and South Africa which have implemented a sugar tax [60,61,62]. Fiscal measures are difficult to implement as they often face strong opposition from powerful industry [63,64]. This is true for Zambia where a sugar tax of 3% was adopted as opposed to the recommended 12% in part due to strong industry lobbying [65,66]. Further strengthening of the sugar tax being implemented Zambia is required to address the negative impacts of the nutrition transition which is underway [27]. However, lobbying for a higher sugar tax should not only be led by the health sector but should garner stronger public support with civil society involvement. Public demand and civil society lobbying were critical in countries that have managed to adopt higher tax rates like South Africa [4,67].

Subsidies to improve affordability of healthy food can also contribute to improved food environments and support healthier choices, as adolescents in our study were shown to have limited purchasing power. Pricing strategies including subsidies on fruit and vegetables have been shown to significantly increase the purchase and consumption of these products in high income contexts [68]. A study in United States of America recorded increased consumption of fruits in adolescents using vouchers in supermarkets that offer subsidies to school-going children [68]. However, the resource constraints of most governments in LMICs might not favor the implementation of such subsidies. One way to support the subsidies would be to use a bundled approach of earmarking revenue from taxation of unhealthy commodities to support healthy food provisioning to critical population groups, which can be carried out through existing nutrition interventions/programs [69]. In Zambia, existing social cash transfers platform can be leveraged to target vulnerable urban adolescents using a voucher system. Public private partnership between retailers and schools can be used for adolescents to redeem vouchers from selected retailers. Such public private partnerships can promote shared responsibility in creating a healthier food supply by stocking, promoting, and reducing costs of healthier foods in their stores [44].

The adolescent’s inability to purchase healthy food due to financial constraints can be overcome by scaling up school feeding programs. School feeding programs have been shown to improve educational and nutritional outcomes of learners, and when coupled with a home-grown component, economical outcomes of communities in many LMICs including Tanzania, India and South Africa have been positively impacted [70,71,72]. Zambia’s school feeding program, like many LMICs, is predominately targeted at learners in lower grades of primary school in rural areas [73]. However, the high rates of urban poverty [21], and inability of adolescents to purchase healthy food as demonstrated by this study provides impetus for expanding the program to urban districts.

### 4.3. Regulating the Marketing of Unhealthy Foods

Our findings also indicate that the adolescents were exposed to food and beverage advertisements through a variety of media platforms like TV, radio, print, and internet. Children and adolescents continue to be the target of marketing activities by the food industry across the globe [13,74,75]. Stricter regulation of unhealthy food marketing to Zambian children via these media platforms is required to minimize the influence of the media on adolescent food choices, as recommended by the WHO [76]. A number of middle- and high-income countries in Europe, Asia and Africa have statutory regulation against marketing of unhealthy food to children [77]. In Chile for example, food advertisements on TV channels popular with children were significantly reduced, following a restriction on food marketing implemented in 2016 [78].

In many LMICs including Thailand and Nepal, implementation and enforcement of marketing regulations remains suboptimal due to lack of political will, evidence to support the regulation and limited capacity to monitor compliance [48,79]. As with other legislative measures like taxation, marketing regulations have been contested in favor of self-regulation by industry, which studies have shown to be ineffective [80,81]. In South Africa for instance, SSB advertisements were still present in most primary schools despite the voluntary pledge made by industry not to advertise their products in schools [82]. Zambia could draw an implementation lesson from the restriction of marketing of breast milk substitutes, which has been in place for over a decade [83], to strengthen the design of marketing restriction of unhealthy food and beverages.

While the policies for regulating marketing are mainly tailored for traditional media such as radio and TV, guidelines on how to regulate social media are still underdeveloped. Regulating the rapidly evolving social media marketing landscape of influencer marketing and user co-created marketing remains a challenge even for high income countries [84,85,86].

### 4.4. Improving Desirability of Healthy Food

Adolescents in our study had a strong desire for unhealthy food which they described as trendy, tasty and appealing as compared to healthy food. Traditional foods, including *nshima*, which is usually eaten by hand, was classified as healthy food but largely seen as unappealing and inconvenient to eat. Modern fast foods like pizzas, burgers, fried chicken, and chips from international franchises like *Hungry Lion* and *Debonairs Pizza* were generally preferred for consumption outside the home. These findings imply the need for interventions to motivate desire for healthy food among adolescents.

The secondary school curriculum in Zambia offers some nutritional education as evidenced by 65% of school-going adolescents who reported having been taught about healthy eating and benefits of consuming fruit and vegetable [52]. School based lessons must be reinforced by social media communication campaigns tailored to adolescents, as these have been shown to improve healthy food habits among adolescents in countries like Indonesia [87,88]. Communication campaigns provide an opportunity to address concerns about affordability of healthy food by promoting consumption of locally available healthy foods that are relatively cheaper. Other communication platforms such as road shows and celebrity endorsement, which we found in this study to appeal to adolescents, should be used to promote the acceptability of healthy food options. In addition, national awareness campaigns promoting healthy dietary habits including healthy food preparation and interpretation of nutritional information are required to support the whole of society’s behavioral change [89]. While education-based policy interventions show a weak positive effect on improving diets [36], they increase population awareness which is required to promote demand and advocacy for more effective nutrition related interventions like taxation and food labelling requirements.

### 4.5. Strengths and Limitation

This study included a large sample size of 200 adolescents from different social economic backgrounds contributing to the credibility of the findings. The approach of triangulating data from FGDs with observations of the school food environment strengthens the trustworthiness of our findings on the food environment factors influencing food choices of adolescents. However, our findings largely represent perspectives of urban school going adolescents and therefore contextual issues should be considered when generalizing to other sub populations of adolescents. In addition, the study only included grade 10 pupils with average of 16 years and therefore the views of younger adolescents might be underrepresented. Future studies are required to understand the food environments and food choices of other adolescent subgroups such as those in the younger age range, rural residents or the out of school, to understand how they compare with in school adolescents included in this study.

## 5. Conclusions

Food environment factors play a role in influencing food choices of school going adolescents in Zambia. Our findings show that the school food environment generally promotes adolescent’s unhealthy food choices due to the wide availability and affordability of unhealthy foods. Food provisioning and retailing polices as well as zoning laws which work to limit the spatial distribution of unhealthy food vendors in the school environment should be considered to improve availability of healthy food. Price related polices such as subsidies on fresh fruits and school feeding programs can contribute to improving access to health food, especially for low social economic status who cannot afford to purchase the relatively expensive healthy food. Regulations restricting marketing of unhealthy food and beverages to children in Zambia are also urgently required. This is because our findings show that adolescents are exposed to food and beverage advertisements via TV, radio, print, and internet channels and such adverts contribute to the desirability of unhealthy foods. Communication campaigns that will empower adolescents and the general public with information to make healthy food choices as well as to demand and advocate for healthy food environments are thus required.

## Figures and Tables

**Figure 1 ijerph-19-07460-f001:**
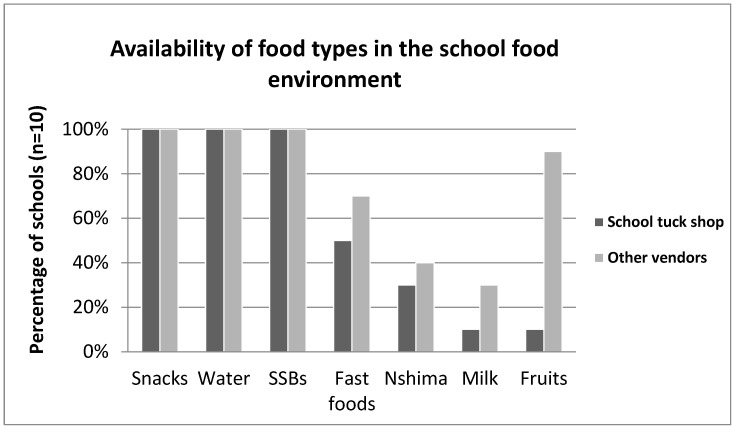
Food availability in the school food environment (Source: School food environment observations).

**Table 1 ijerph-19-07460-t001:** Secondary schools in Lusaka district included in the study.

School Code	Population Density of School Neighborhood	Ownership
1	High density	Government
2	High Density	Government
3	High Density	Government
4	Medium density	Government
5	Medium density	Government
6	Medium density	Government
7	Low density	Government
8	Low density	Government
9	Low density	Government
10	Low density	Private

**Table 2 ijerph-19-07460-t002:** Codes, subthemes and themes used in data analysis.

Example of Codes	Sub Themes	Themes (Based on the Conceptual Framework)
Healthy food does not look and taste appealing Appetite and cravings determine what you eat Experimenting with different food Eating outside the home is a chance to eat other things aside from the monotonous staple food Fast food is trendy among adolescents Fast foods like hungry lion and burgers look appealing Young people like food that gives energy because they are very active Young people like food that is enjoyable to eat Eating healthy food like fruit and salad makes one unique	Taste and preference	Desirability
Each household has what they eat Parents are role models of what to eat	Culture	Desirability
Healthy food provides nutrients for good health Junk food has no nutrients Modern food makes people sick Fast food reduces appetite for healthier food Food preparation method determines whether food is healthy or not Quantity of food consumed will determine whether it is healthy or not Junk food is related to diseases like diabetes and dental caries Fruits eaten when one is unwell Traditional food is healthy Healthy food is eaten at home	Nutrition related knowledge	Desirability
Young people cannot afford healthy food like fruits Young people do not have much money Healthy food requires a lot of ingredients		Affordability
Fruits are sold very far from school		Accessibility
Young people do not have time to prepare traditional food Traditional food is messy to eat Healthy food takes long to prepare Young people like food that is easy to cook Noodles and bread are easy to prepare by adolescents		Convenience
Healthy food costs more money than junk food		Price
Cheap junk food is readily available at school Fruits and other healthy food are not sold by school tuck shops Vendors only sell what young people can afford Fruit trees are available at the school		Availability
Some adverts for soft drinks are targeted at young people as they use songs by celebrities that appeal to young people Adverts make young people start consuming fast foods There are a lot of adverts for sugar sweetened beverages on TV and billboards Social media has adverts for soft drinks Celebrities promote food and drinks		Marketing
Hygiene of the vendor premises is very important Presence of hand washing station for preventing COVID Location of the vendor Packaging for food items should be clean Food packaging should have expiry date Vendor should be popular with other young people		Vendor and product properties

## Data Availability

Not applicable.

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
