# Peer review of "School Food Environment in Urban Zambia: A Qualitative Analysis of Drivers of Adolescent Food Choices and Their Policy Implications"

_ijerph, 2022, doi:10.3390/ijerph19127460_

Round 1
Reviewer 1 Report
- Line 117, it is mentioned that purposive sampling was done. There is need to explain the reason for choosing purposive sampling over random sampling.
- Regarding Drivers of Food Choice, starting line 208, it would be more scientific if multilinear regression is performed on the collected data to identify these drivers of food choices.
Reviewer 2 Report
Journal: International journal of Environmental Research and Public Health
Manuscript ID: ijerph-1647480
Title: School Food Environment in Urban Zambia: An Analysis of Drivers of Adolescent Food Choices and Their
Policy Implications
Brief Summary
The aim of this qualitative study was to identify the food environment factors influencing food choices of school
going adolescents in Zambia. Researchers conducted focus group discussions and school food environment
mapping of 10 secondary schools in urban Lusaka. Results showed that internal drivers of food choice included
desirability, affordability, accessibility, and convenience. External drivers of food choice were price, availability,
and marketing. Conclusions highlighted the need for policy reforms and regulations to improve the food school
environment and support healthy food choices in adolescents.
General comments
• This study addresses a gap in the knowledge about key determinants of adolescent food choices in the
food school environment in urban Zambia.
• While overall a well-written and interesting paper, the authors are advised to refer to the Standards for
Reporting Qualitative Research (SRQR) in revising this manuscript for rereview. The SRQR can be found on
the EQUATOR network website at https://www.equator-network.org/reporting-guidelines/srqr/.
• It is also recommended that the authors attempt to complete the SRQR checklist to ensure all the
required elements are included in the manuscript. A detailed description of each reporting item and
examples are available in Supplemental Digital Appendix 2 at: https://links.lww.com/ACADMED/A218
This will be very helpful in revising the manuscript, especially the Methods section.
• Many references did not match the statement they followed so should be replaced.
Specific comments
Title
Suggest revising the title to give information about what type of study this was, i.e., that this study was qualitative
or that the method was semi-structured interviews. For example, “School Food Environment in Urban Zambia: A
Qualitative Analysis of Drivers of Adolescent Food Choices and Their Policy Implications”
Introduction
Line 37: Please replace this reference with one or more that match the statement, “social cognitive factors...have
globally been shown...”. The cited reference only reports a study done in Ireland. In addition, it would be better
to use a more recently published reference.
Lines 41-43: This sentence doesn’t make sense as written. Are food consumption and overall nutritional
outcomes community level influencers of food choice? Isn’t more that they are the result of food choice?
Line 43: References #6 and 7 are both books. If specific pages were referenced, that information should be
included in the reference.
Lines 53-57: The two sentences starting with “The school food environment ...” are verbatim from reference #14.
To avoid plagiarism either rewrite in other wording or leave as is and add quotation marks.
Lines 63-65: Regarding reference #19, same comment as above for Lines 53-57. Plagiarism.
Line 77-78: The authors write that “adoption and implementation...remains low in many African contexts”, but
the paper cited refers only to Ghana. Please add other citations to back up the “many African contexts”
statement.
Lines 86-88: Reference #26 is listed in the text as unpublished and published online. Which is it? If it is published
ahead of press, online information should be added to the reference so the reader may access the article.
Materials and Methods
General comment: Please read the SRQR Supplementary Appendix 2 detailed description of each item and
accompanying explanations and examples. Use that information as a guide to add items currently missing from
the Methods, for example:
• When was the study conducted? Please provide beginning and end dates (month, year).
• Please add the following required language from IJERPH Instructions for authors “ “The study was conducted
according to the guidelines of the Declaration of Helsinki and approved by the Institutional Review Board (or Ethics
Committee) of NAME OF INSTITUTE (protocol code XXX and date of approval).”
• Were participants compensated?
Line 117-121: How was the sample size of 10 schools determined?
• Who decided on which schools to approach? Please describe the purposive sampling procedure in detail.
• How many schools were in each category (high, medium, and low density)?
• The authors mention that private schools were also approached but only one granted permission. How
many were approached? Were there any public schools that declined? This information could be here or
in the Results section.
Lines 118-120: Please explain how the high, medium, and low density residential areas were a proxy for different
socioeconomic and economic status. Which residential area corresponded with higher or lower income? While
clear to the authors, it is not clear to the reader.
Lines 120-121: If only one school was a private school, be specific here what type of school the other nine were. It
is mentioned later on Line 138 that they were government funded schools, but it should be mentioned here first
for clarity.
Line 123: Above on Line 112 the authors write that there are two domains to Turner’s framework but here it is
listed as having eight domains. There are eight dimensions in Turner’s framework so that word should be used.
Line 123 and Line 128: Please provide a table with the questions in the semi-structured guide for the FGDs and
another for the semi-structured observation checklist.
Line 130: How were the observations recorded? By taking notes, photographs, video? Who conducted the
Line 135: Please specify how the random selection of participants was done and by whom (teachers, researchers).
• Describe the consenting/assenting process of the students
Line 136-137: It is not sufficient to state that the sampling and recruitment details are available from the
corresponding author. That information should be included in this manuscript.
Lines 139-140: Please describe “quiet environment” and did it differ among schools?
• On average, how long did the FGDs last?
• See Item 13 in SRQR (data processing) and address all points
Line 141: How many research assistants? Please describe their training.
Line 154: Where is Table 1?
Line 164: Supplementary Table 1 should be moved to the body of the manuscript as it is important for the reader
to see.
Results
Were there any of the selected students who declined or did not compete any FGDs?
Discussion
Line 409: Reference #46 is from a study on LMICs, not the USA. Please correct.
Line 413: Not sure why a single study from Nepal is referenced here when the authors are talking about “most
LMICs including Zambia”. A better reference is needed.
Line 426: Please remove reference #50 as it does not discuss improved nutritional outcomes of children.
Line 431: Regarding reference #52, same comment as for Lines 86-88.
Lines 437-439: Missing a reference for South Africa
Lines 466-469: The sentence beginning with “School feeding programs...” is incomplete.
Lines 492-493: Revise “breast milk supplements” to “breast milk substitutes”.
Line 518: The reference #82 is about tobacco packaging laws, so does not fit the statement. Since the statement
refers to the current study’s findings, no reference is needed here.
Line 521: Reference #83 is not about national awareness campaigns, but about interventions at retail stores and
supermarkets. Please replace with a fitting reference.
Round 2
Reviewer 1 Report
I am not sure about response 4. Including multilevel modelling was going to make the paper stronger rather than splitting the paper into two, one using qualitative data and another using quantitative.
Usually, qualitative data is used to support findings and trends revealed from quantitative data.
